



# The MArine Debris hyperspectral reference Library collection (MADLib)

Ashley Ohall[1], Kelsey Bisson[2], Shungudzemwoyo P. Garaba[3] and Sara Rivero-Calle[1]

[1]Skidaway Institute of Oceanography and Department of Marine Sciences, University of Georgia, 10 Ocean Science Circle, Savannah, GA 31411, USA
[2]Earth Science Division, Ocean Biology and Biogeochemistry Program, National Aeronautics and Space Administration Headquarters, Washington, DC, USA
[3]Niedersächsische Zentrum für Marine Sensorik, Carl von Ossietzky Universität Oldenburg, Schleusenstraße 1, 26382 Wilhelmshaven, Germany

*Correspondence to*: Sara Rivero-Calle (rivero@uga.edu), Shungudzemw Garaba (shungu.garaba@uni-oldenburg.de)

**Abstract.** Marine debris is a ubiquitous and growing threat to environmental and human health. Efforts to monitor and mitigate marine debris pollution face many challenges. A primary limitation is the absence of standardized methodologies for monitoring capabilities due to the complex and diverse physical and chemical properties of marine debris. Variabilities include object size, apparent color, polymer type, weathering, and aqueous state. Despite the challenges in object characteristics, advances in remote sensing methods are showing promise for detecting marine debris across local to global scales. Algorithms are needed to link remotely sensed observations with relevant characteristics of marine debris to fully realize this potential. Although more optical measurements of marine debris reflectance are becoming available for algorithm development, inconsistencies in data curation remains an obstacle. Variations in data processing and inconsistent metadata hinder efforts to develop robust, generalizable algorithms for marine debris detection. To address this, we present the well-curated MArine Debris hyperspectral reference Library collection (MADLib) containing 24889 spectra from 3032 samples. All optical measurements are available in open access via https://doi.org/10.4121/059551d3-2383-4e20-af2d-011c9a59d3ac (Ohall et al., 2025). MADLib demonstrates the importance of open-science and open-access datasets, as it compiles and harmonizes spectral data collected from publicly accessible datasets and individual research projects. Consistent methods were applied for data standardization, quality assurance, and integration. We also propose a robust protocol for generating metadata tailored to marine debris and ocean color remote sensing applications. MADLib possesses spectra of a wide range of marine debris materials including different polymer types, color, size, weathering, and aqueous states. Here, we analyze the metadata associated with the spectra to identify sampling gaps and propose considerations for future work. By providing open-access and standardized data, MADLib is expected to support the development of robust marine debris detection algorithms.

**Keywords**: Marine debris; Hyperspectral reflectance library; Plastics; Polymer.





## 1. Introduction


Marine debris or litter is any persistent solid material that is manufactured or processed and directly or indirectly
disposed of or abandoned in an aquatic environment (Cheshire et al., 2009). Marine debris has become ubiquitous
across all aquatic environments due to the rapid production of manufactured goods without proper disposal
management (UNEP, 2021; Thompson et al., 2024; Galgani et al., 2025). The negative implications of mismanaged
marine debris on blue economic activities, environmental and human health are extensive, prompting a need for
effective monitoring and tracking to support informed mitigation strategies (Beaumont et al., 2019; Smith and Garaba,
2025; GIZ, 2023; NASEM, 2021).
Remote sensing has the potential to support the monitoring of aquatic debris concentrations and dispersal patterns
across spatial and temporal scales. From this perspective, the definition of marine debris is broadened to include not
only anthropogenic materials, but also natural materials such as wood, pollen, pumice or seaweed species (Martínez-
Vicente et al., 2019; Maximenko et al., 2019; NASEM, 2021; Hu et al., 2023). However, ongoing efforts in remote
sensing of marine debris are challenging due to the complexity of the targets (de Vries et al., 2023a; NASEM, 2021;
GIZ, 2023). Debris objects vary widely in physical properties (e.g., color, shape, composition, size) that are further
influenced by environmental factors such as aqueous state and stage of weathering (**Figure 1**). Fully understanding

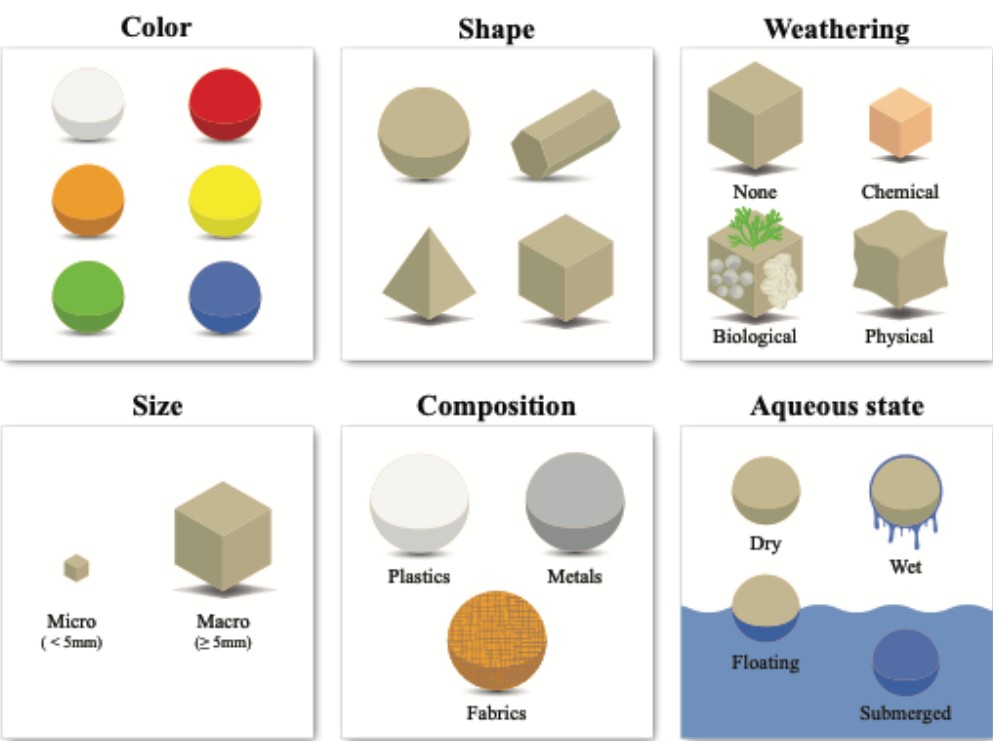

**Figure 1. Examples of physical and chemical characteristics of marine debris in nature.**



marine debris diagnostic optical properties is essential for building, training, and validating remote sensing detection algorithms (Garaba et al., 2021a). This information, together with the evolution of hyperspectral remote sensing technologies and the development of active and passive sensors, is expected to further advance current capabilities to detect and monitor marine debris.

The reflectance parameter is one of the common remote sensing parameters that describes the ratio of light reflected off an optically active sample with respect to a known standard like a Lambertian equivalent target. Reflectance measurements, when collected under controlled conditions using instruments such as handheld spectroradiometers, minimize the environmental variability, which is ideal for algorithm development (Knaeps et al., 2021; de Vries et al., 2023a; Garaba et al., 2021a). Recent stakeholder discussions facilitated by the International Ocean Color Coordinating Group Task Force on Remote Sensing of Marine Litter and Debris highlighted the need for a comprehensive, well-curated spectral reference library (SRL) of marine debris reflectance. SRLs support algorithm development by offering spectral data of known objects, establishing a baseline database that can be used to identify unknown objects. However, current marine debris reflectance datasets were not designed for interoperability, and their inconsistent formatting makes it challenging to combine them for algorithm development and to identify gaps in the field.

In this study, we leverage the wealth of available spectral reflectance measurements of marine debris to build a consistent and extensive collection. We compiled, assessed and curated the available marine debris reflectance datasets into a single SRL called the MArine Debris hyperspectral reference Library collection (MADLib). MADLib aims to improve the accessibility and comparability of current data to promote the spectral exploration and analysis needed for marine debris algorithm development. We also aimed to follow the FAIR guidelines by making the data findable, accessible, interoperable, and reusable (Wilkinson et al., 2016). Here, we explain how the data contained in the collection was curated, identifying existing sampling gaps, and discussing factors that are important for the success of future SRLs and remote sensing of marine debris.

## 2. Methods and materials

### 2.1 Selection of datasets

Thirteen datasets were selected for curation and creation of MADLib collection from open-access sources as well as upon request from authors (**Table 1**). The selected datasets have the following characteristics (i) the data reported were relative reflectance not remote sensing reflectance, (ii) reflectance was measured using a handheld spectroradiometer, and (iii) hyperspectral data were provided in the visible to Shortwave Infrared (SWIR) region. The remaining datasets either did not meet these criteria, were not readily available upon request, or needed further curation from the authors (e.g., Acuña-Ruz and Mattar, 2020; Olyaei et al., 2024; Tasseron et al., 2021; Wang et al., 2024; Knaeps et al., 2020). Leveraging the datasets (Table 1), we determined a standard formatting structure from which to build the MADLib collection. This included several spectral data processing steps for quality control (Section 2.3) and gathering additional metadata parameters about the samples (Section 2.4).



**Table 1. Source of hyperspectral measurements used to create MADLib.**

| Dataset Number | Reference | Data access | Number of samples | Keywords |
|---|---|---|---|---|
| 1 | (Corbari et al., 2020) | Author Permission | 65 | Dry, Floating, Pristine, Micro, Varying Thickness, Varying Pixel Coverage |
| 2 | (de Vries and Garaba, 2023) | CC BY 4.0 | 575 | Dry, Wet, Submerged, Pristine, Naturally Weathered |
| 3 | (de Vries et al., 2023b) | CC BY 4.0 | 115 | Dry, Submerged, Lab Weathered |
| 4 | (English and Hu, 2020) | ODC BY 1.0 | 6 | Dry, Floating, Pristine, Naturally Weathered |
| 5 | (Garaba et al., 2021a) | CC BY 4.0 | 793 | Dry, Floating, Pristine, Naturally Weathered, Varying Pixel Coverage |
| 6 | (Garaba et al., 2020) | CC BY 4.0 | 80 | Dry, Wet, Submerged, Pristine, TSM |
| 7 | (Garaba and Dierssen, 2017) | CC BY 4.0 | 11 | Dry, Pristine, Micro |
| 8 | (Garaba and Dierssen, 2019b) | CC BY 4.0 | 2 | Dry, Wet, Naturally Weathered, Micro |
| 9 | (Garaba and Dierssen, 2019c) | CC BY 4.0 | 6 | Dry, Naturally Weathered, Micro (specific size classes) |
| 10 | (Garaba and Dierssen, 2019a) | CC BY 4.0 | 23 | Dry, Naturally Weathered, Macro |
| 11 | (Garaba et al., 2021b) | CC BY 4.0 | 9 | Dry, Floating, Naturally Weathered, Pristine |
| 12 | (Leone et al., 2021) | CC BY 4.0 | 1077 | Dry, Wet, Submerged, Pristine, Lab Weathered, Naturally Weathered, TSM, Algae |
| 13 | (Corbari et al., 2024) | Author Permission | 270 | Naturally Weathered, Black Background vs White Background |


**2.2 Materials**
MADLib includes 3032 samples compiled from thirteen datasets (**Table 1**). Each sample represents either a single
marine debris object (e.g., a bottle or buoy) or an assemblage of micro-sized items (e.g., a collection of microplastic
particles measured together) measured under specific conditions. For example, the same object measured in both dry





and submerged states was represented as two separate samples in MADLib. Each sample is associated with a spectrum
and the related metadata. The samples encompass a wide variety of colors, sizes, polymer types, weathering
conditions, aqueous states, and experimental designs. It should be noted that, while these datasets include various
debris types, plastic is the dominating type of debris reported, reflecting its overwhelming presence in the marine
environment.

## 2.3 Spectral data processing

Each dataset was downloaded from its respective open-access platform or requested from the corresponding author
and assembled into MADLib. Spectral reflectance measurements covered a wavelength range between 280-2500 nm
with 1 nm resolution. In most cases, multiple spectral measurements were recorded per sample, sometimes in various
geometric orientations. We will refer to these measurements as replicates, which account for within-sample variability
and instrument noise.
The final MADLib spectral data download includes five identification columns: *dataset number, sample number, data
type, replicates* and *flags*. *Dataset number* refers to the cited datasets (**Table 1**). *Sample number* uniquely identifies
each sample within a specific dataset. *Data type* specifies whether the data in that row represents the "mean", "median",
or standard deviation "stdev" of that sample's replicates, or "single" for single measurements without replicates.
*Replicates* provides the number of replicates associated with that statistical representation. *Flags* assigns a "1" to
spectra containing more than 50 % NaN values and otherwise assigns a "0". The data are sorted alphanumerically by
*data type*, then *dataset number,* and finally *sample number*.

### 2.3.1 Data formatting

Spectral data were obtained in one of four formats depending on the dataset: (1) individual spectral measurements for
each replicate, (2) pre-calculated means, medians, and standard deviations of the replicates per sample, (3) only the
mean spectral reflectance values of the replicates per sample, or (4) single reflectance measurements per sample
without replicates.
When more than one individual spectral measurement per sample was provided, the mean, median, and standard
deviation of the replicate measurements were calculated for each sample to standardize across datasets. Two datasets
provided only mean spectral reflectance values of their sample's replicate measurements, so the median and standard
deviation for these data were written as NaNs (Not a Number) in MADLib. Single measurements were provided for
six samples across three datasets and were classified separately as "single".
MADLib only reports the descriptive statistics of the compiled data as mean, median and standard deviation of the
replicates (with the two exceptions specified above). In total, MADLib summarizes the information from 24889
replicate measurements of 3032 samples collected from thirteen datasets (**Figure 2**). In the datasheet, 3026 mean
measurements, 2691 median measurements, and 2691 standard deviation measurements are recorded. Six samples did
not have replicates, so they are available as individual measurements without summary statistics.

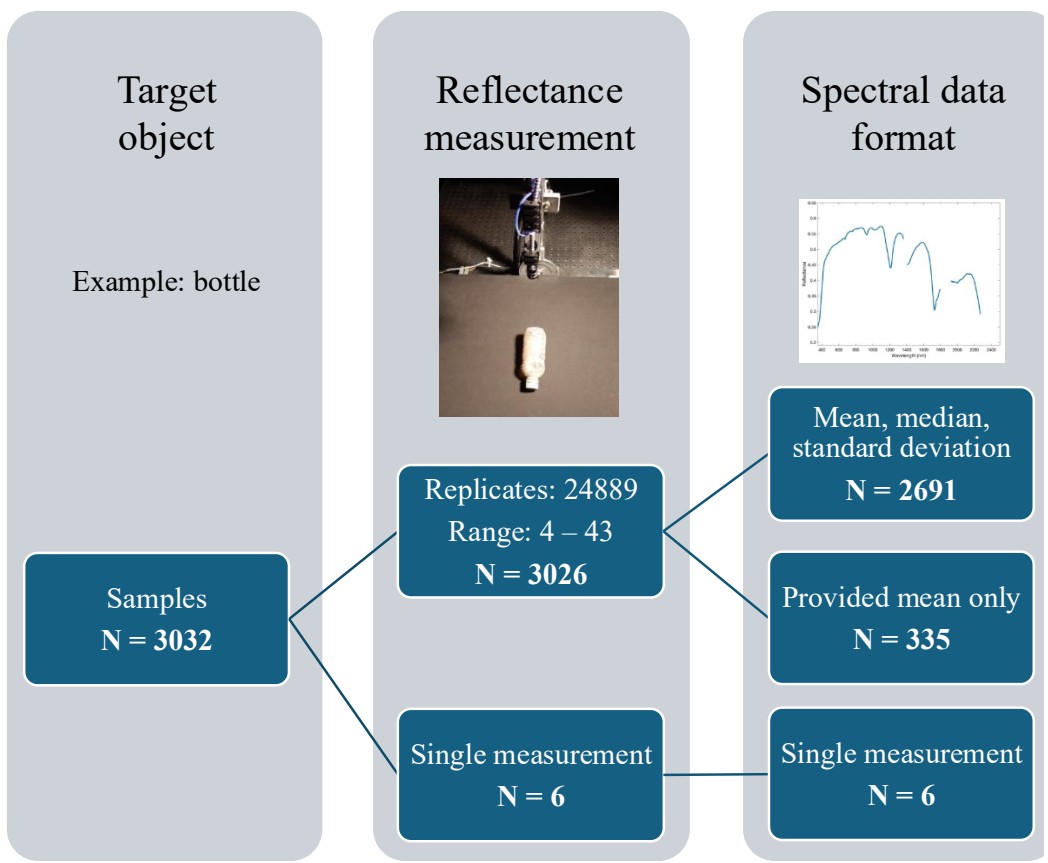

**Figure 2. Breakdown of the number of samples, replicate measurements for total samples, and descriptive statistics of sample spectra within MADLib.**


**2.3.2 Wavelength range adjustment**
A wavelength range limit of 280-2500 nm was applied to ensure consistency across datasets. NaNs were used in place
of the missing spectral data for instruments not collecting data in the fixed range.
**2.3.3 Splice correction**



Hyperspectral instruments measuring beyond the visible-near-infrared (VNIR, 280-1000 nm) consist of multiple
detectors, each covering a distinct spectral range. Off-the-shelf spectroradiometers commonly used in environmental
remote sensing applications (e.g., Analytical Spectral Devices FieldSpec 4, Spectral Evolution SR-3501, Spectral
Evolution SR-1901) have three detectors. When transitioning between the detectors, slight differences in sensitivity,
temperature or calibration can create discontinuities in the reflectance spectral measurements. The spectral
discontinuities, or "steps", usually occur around 1000-1001 nm and 1800-1801 nm, but the exact positions are
instrument and manufacturer specific (**Figure 3a**). The spectral data from each dataset were visually inspected for
steps. If a step was identified, we calculated the linear difference at each step and adjusted one region of the spectrum
based on another to eliminate the gap (Garaba et al., 2021a). The middle detector (1000-1800 nm) was considered as
the reference, and the adjacent regions (280-1000 nm and 1800-2500 nm) were adjusted to that reference level. For
example, if the reflectance difference between 1000 nm and 1001 nm is -0.02, then 0.02 is subtracted from all values
in the 280-1000 nm range to align it with the more stable middle region (**Figure 3b**).

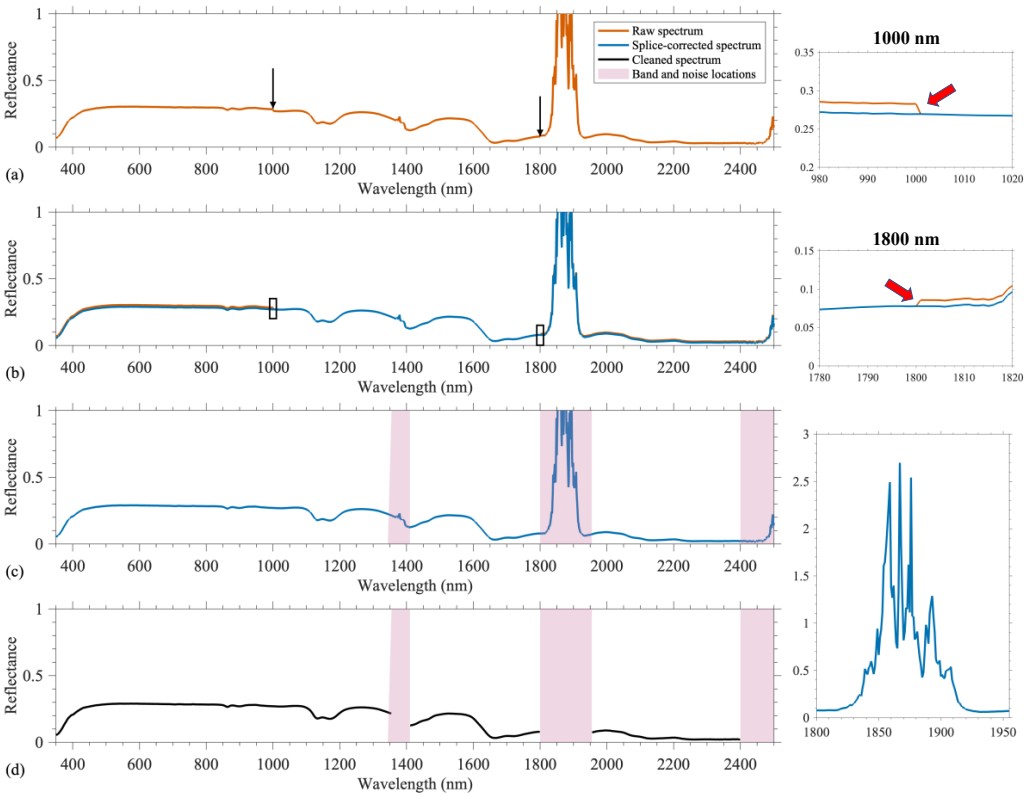

**Figure 3. Example processing steps for spectra: (a) raw downloaded spectrum; (b) comparison of raw and splice-corrected spectra, (c) identification of atmospheric absorption bands and instrument noise; and (d) final cleaned spectrum. Zoom-in boxes show the 980–1020 nm and 1780–1820 nm regions in (b), highlighting the steps and splice correction at 1000 nm and 1800 nm, and the 1800-1950 nm region in (d), highlighting the full reflectance magnitude of the atmospheric absorption band.**





### 2.3.4 Noise removal

Visual inspection was used to identify, and subsequently remove, noise in the spectra. The affected wavelengths were replaced with NaNs to avoid misinterpreting them as real spectral features in later analyses (**Figure 3d**). Noise was considered to arise from two main sources: atmospheric absorption bands and instrument-related noise. Atmospheric absorption occurs in regions where the atmosphere is opaque, specifically around 1350–1450 nm, 1800–1950 nm, and above 2400 nm (Garaba and Dierssen, 2020; Clark et al., 2003). This is particularly evident in outdoor measurements, where spectra often exhibit abrupt, isolated peaks due to these absorption features. Instrument noise was also observed, particularly at the extreme ends of the spectral range, where sensor sensitivity tends to decrease.

### 2.3.5 Flags

Hyperspectral data (mean, median, standard deviation, or single measurement) with more than 50 % NaN values across the original wavelength range were flagged. The flagged entries were kept in MADLib for completeness but were marked with an additional *flags* column to indicate data quality. A binary code of "1" indicates a flagged sample, while "0" indicates a clean sample.

### 2.4 Metadata curation

MADLib incorporates the unique metadata provided for samples from each dataset, adds new metadata parameters, and ultimately provides comprehensive metadata descriptors for improved interoperability. Metadata descriptors date/time, longitude, latitude, FTIR identification, and sample weight were excluded from the curation due to their limited applicability across datasets and the potential for misleading interpretation since the samples were not imaged in situ. Metadata were incorporated from existing metadata files, descriptions within associated publications, and, when necessary, missing details were obtained from the authors directly. When this was not possible, NaN values were assigned to indicate the missing metadata. The final MADLib metadata download includes thirty-nine columns of metadata descriptors **(Table 2)**. The thorough curation process in MADLib enabled more detailed and robust analyses, focusing on parameters that enhance the identification and classification of marine debris through reflectance measurements.

**Table 2. List of MADLib metadata descriptors.**

| Meta column name | Description |
| --- | --- |
| DatasetNumber | Unique library identifier |
| SampleNumber | Unique sample identifier |
| Polymer Type | Standard abbreviation for plastic polymer (**Table 3**) |



| Object Type | Purpose or use of object |
|---|---|
| Object State | Describes how an object was physically altered or interacted with |
| Origin | Manufacturer or location sample was collected |
| White / Transparent / Red / Orange / Yellow / Green / Blue / Purple / Brown / Gray / Black / Multi | Binary indicator for the apparent color of sample: present (1) or absent (0) |
| Length | First dimension provided (mm) |
| Width | The second dimension provided, if applicable (mm) |
| Height | The third dimension provided, if applicable, or sample thickness (mm) |
| Categorical Size | Micro or macro |
| Weathering State | Pristine, lab weathered, or naturally weathered |
| Laboratory Weathering Type | If lab weathered: phytoplankton/biofilm, UV degradation, or other |
| Aqueous State | Dry, wet, submerged, or floating |
| Submergence Depth | Sample depth below water surface (mm) |
| Water Type | Freshwater, saltwater, seawater, artificial seawater, or filtered seawater |
| TSM | Total suspended matter concentration (mg/L) |
| Algal Cell Density | Number of microalgae cells per milliliter of water (cells/mL) |
| Pixel Coverage | Proportion of instrument field of view covered by an object (%) |
| Glass Presence | G = glass held sample in place, N = no glass, NaN = not applicable |
| Setting | Indoors or outdoors |
| Instrument | Manufacturer or brand name of spectroradiometer |
| Lighting | Artificial or ambient natural light source |
| Background | Black background, white background, concrete, land, water |




| Reference Standard | Reference plaque reflectance percentage |
|---|---|
| Fixed Height from Sample | Distance from fore optic to object surface (dry, wet) or water surface (submerged, floating) (m) |
| FoV | Field of view of the bare fiber optic or fore optic lens (deg) |
| Viewing Geometry | Nadir viewing angle (deg) |


### 2.4.1 Sample type

Despite MADLib containing a variety of marine debris types, it is primarily composed of plastic debris. Consequently,
this paper will address plastic-specific characteristics, such as polymer type, in addition to broader characteristics.
Sample type parameters included *polymer type, object type, object state,* and *origin* to maximize dataset comparability.
For example, if a dataset described a sample as "crushed PET water bottle", we further described it using *polymer type*
= "PET", *object type* = "bottle", and *object state* = "crushed". During this process, the *polymer type, object type*, and
*object state* were simplified or modified to ensure consistency and comparability across datasets. For example, object
types labeled as "water bottle", "clear water bottle", "bottle", "plastic bottle", or any similar terminology were all
simplified to "bottle". *Origin* was provided if the dataset specified the manufacturer or place the item was retrieved
from.

### 2.4.2 Color

Colors are categorized as *white, transparent, red, orange, yellow, green, blue, purple, brown, gray, black,* and *multi*
to ensure consistency and comparability across samples. For example, pre-existing samples marked as tan, dark brown,
light brown, and brown were all included in the *brown* category. Colors were recorded with binary entries to easily
identify objects with multiple colors. If more than one color was specified by the original author, all relevant colors
were marked with a 1. If multi-colored was specified by the original author, only *multi* was marked with a 1.

### 2.4.3 Size

Object dimensions were recorded differently across datasets and required representation in various formats. The
*length, width,* and *height* columns were used for objects with complete dimensional data, while the *height* column
additionally represented thickness where relevant. If a range of sizes was provided (e.g., 1-3 mm) by the authors of
the original dataset, then the average was included (e.g., 2 mm) in MADLib. If a height or thickness of <1 mm was
provided, then 1 mm was reported. In some cases, the authors alternatively provided categorical size data, either
"micro" (<5 mm) or "macro" (>5 mm), so we included a *categorical size* classification. Samples with numerical data
provided were categorized as "micro" if all three dimensions (L, W, H) were <5 mm. If only partial size data were



available and under 5 mm, samples were not categorized as "micro" to avoid errors in cases where a missing dimension might exceed 5 mm. Conversely, any sample with one or more dimensions >5 mm was classified as "macro".

### 2.4.4 Weathering

*Weathering state* was specified as either "pristine", indicating non-weathered, virgin material; "lab weathered", subjected to controlled laboratory conditions; or "naturally weathered", collected from the marine environment and exposed to natural processes. The specific type of lab weathering was indicated as either "biofouled" or "UV exposure" in the *lab weathering type* metadata column. Samples labeled as "biofouled" were submerged in natural water within a mesocosm to promote biofilm growth (de Vries et al., 2023b; Leone et al., 2021). Some of these samples were additionally labeled as rough if their surfaces had been abraded with sandpaper prior to submersion (Leone et al., 2023). To simulate photodegradation, other samples were exposed to ultraviolet (UV) radiation under either dry or wet conditions (Leone et al., 2023).

### 2.4.5 Aqueous state and water properties

Four categories describe the a*queous state* of the samples: "dry", "floating", "submerged", and "wet". "Dry" samples refer to dry objects measured on a dry surface. "Wet" samples refer to wet objects measured on a dry surface or above a water body. "Floating" samples refer to any object floating on the surface of a water body or in a water tank. "Submerged" samples refer to any object where the top is at least 1 mm under the water's surface.

If the sample was categorized as "wet", "floating", or "submerged", and information on the properties of the water in which it was measured were available, they were also included. *Water type* specifies if the sample was measured in "freshwater", "saltwater", "seawater" (unfiltered), "artificial seawater", or "filtered seawater". In some cases, samples were measured in a mesocosm or water bath that had added total suspended matter (TSM) or phytoplankton (Leone et al., 2021; Garaba et al., 2020). Concentrations were included in *TSM* and *algal cell density* columns.

### 2.4.6 Experimental setup

The location of measurement, *setting*, was categorized as "indoors" or "outdoors" for all samples. *Lighting* was similarly categorized for each location, with indoors using tungsten halogen lamps, or outdoors using sunlight with recorded conditions (**Figure 4**). Studies also varied with the *viewing geometry, field of view (FoV)*, and *fixed height from sample*. Dry surface and water bath samples were measured on a black background with the exception of one dataset which used both black and white backgrounds for comparison (Corbari et al., 2024).



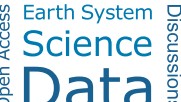
Other controlled sample parameters included *Pixel Coverage* and *Glass Presence. Pixel Coverage,* also referred to as
areal fractional cover, refers to the percentage of the field of view containing the sample and was used to measure
varying concentrations of microplastics (Garaba et al., 2021a; Corbari et al., 2020). Glass was used in one dataset to
hold samples in place, therefore causing a possible disruption to the produced spectra (de Vries and Garaba, 2023).

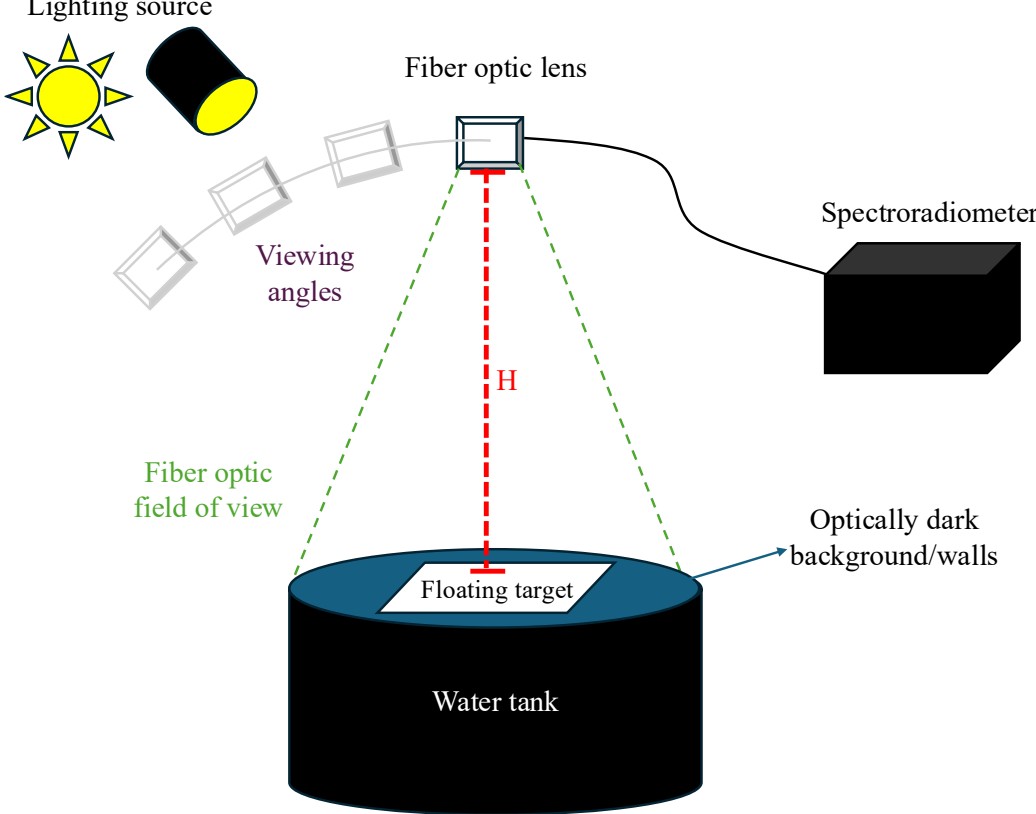

**Figure 4. Schematic of typical experimental setup with the light source, variable viewing geometry, fiber optic field of view, fixed height from sample (H), and an optically dark background.**

**Data availability**
MADLib is available in open access via https://doi.org/10.4121/059551d3-2383-4e20-af2d-011c9a59d3ac (Ohall et
al., 2025). Two CSV files are included with the MADLib download: a metadata sheet and a data sheet. The samples
can be linked across the two files using the *dataset number* and *sample number* columns.
**3. Results**
Here, we examine the distributions of several characteristics within MADLib and present case studies of spectral
reflectance where relevant.



### 3.1 Polymer type

Nineteen distinct polymer types are included in the dataset (**Table 3**). The largest group of samples are of an unknown polymer type (35 %) (**Figure 5a**). The unknown polymer category includes plastics with unidentified polymer types as well as non-plastic marine debris, such as fabrics, metals, and background materials. Polypropylene (PP) is the most common polymer type, making up 15 % of the samples, followed by polystyrene (PS) and high-density polyethylene (HDPE) (**Figure 5a**). Only one sample is available for each of the following six polymers: terpolymer lustran 752 (ABS), fluorinated ethylene propylene teflon (FEP), merlon, polyamide 6.6 (PA6.6), polymethyl methacrylate (PMMA) and thermoplastic elastomer (TPE). Some polymer types in MADLib exhibit similar spectral features across the NIR–SWIR range, while others display distinct or minimal spectral features (**Figure 5b**).

To illustrate MADLib's potential for detailed examination of spectral features, the reflectance spectra of all dry PP and HDPE samples were isolated and presented separately (**Figure 5c, d**). The absorption features are consistent across samples of the same polymer type and align closely with reported literature (Olyaei et al., 2024; Garaba and Dierssen, 2020).

**Table 3. Standard abbreviations of polymer types in MADLib.**

| Abbreviation | Polymer |
| --- | --- |
| ABS | Acrylonitrile butadiene styrene (lustran 752) |
| EVA | Ethylene vinyl acetate |
| FEP | Fluorinated ethylene propylene teflon |
| HDPE | High-density polyethylene |
| HDPE_LDPE | A combination of high and low-density polyethylene |
| LDPE | Low-density polyethylene |
| Merlon | Merlon |
| PA6 | Polyamide 6 (nylon 6) |
| PA6.6 | Polyamide 6.6 (nylon 6.6) |
| PE | Polyethylene |
| PET | Polyethylene terephthalate |





| PETa | Polyethylene terephthalate - amorphous |
|------|----------------------------------------|
| PETc | Polyethylene terephthalate - crystalline |
| PMMA | Polymethyl methacrylate |
| PP | Polypropylene |
| PS | Polystyrene |
| PS-XT | Extruded polystyrene |
| PVC | Polyvinyl chloride |
| TPE | Thermoplastic elastomer |

241

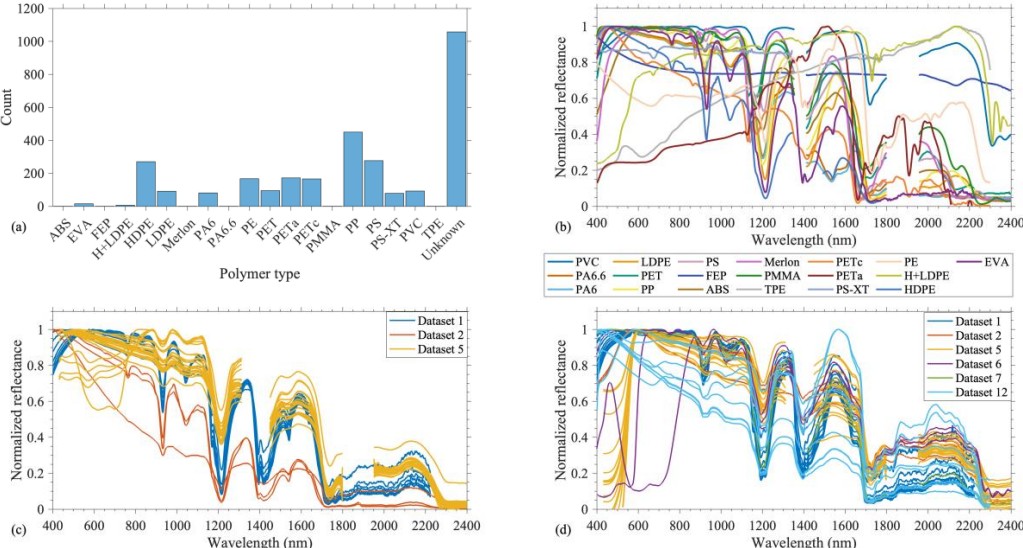

**Figure 5. (a) Distribution of polymer types; (b) representative mean reflectance spectra of each polymer type; (c) mean reflectance spectra of all available dry HDPE samples; and (d) mean reflectance spectra of all available dry PP samples. All reflectance spectra were normalized to their respective maximum values.**

242



## 3.2 Object type

Thirty-nine object types are used to define the samples, noting that 17 % of samples are of unknown object type (**Figure 6**). Among those that could be classified, the top two categories are sheet (33 %) and manufactured plate (16 %). We note that the majority (88 %) of samples labeled as sheet and all samples labeled as manufactured plate were obtained directly from the manufacturer as a single polymer composition. There are five object types (bubble wrap, cloth, lid, sweater, and tire) for which only one sample was measured.

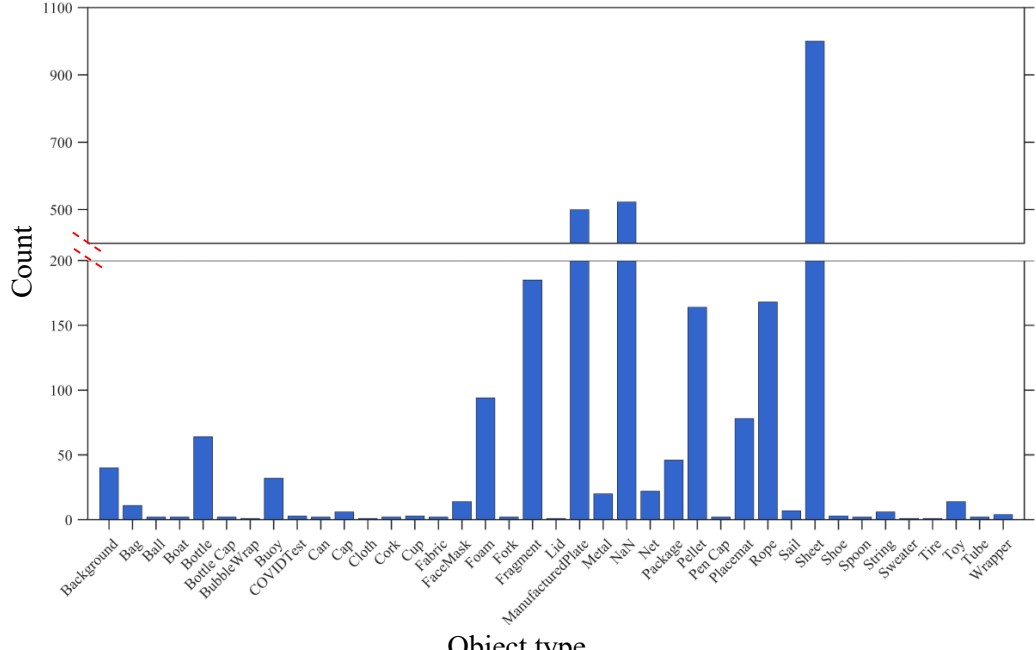

**Figure 6. Distribution of object types among samples in MADLib.**

## 3.3 Color

Twelve color categories are used in MADLib. The three largest categories are unspecified (30 %), white (19 %), and grey (15 %) (**Figure 7a**). The categories brown, black, blue, and transparent contain approximately 5-8 % of the total samples each. 16 % of the samples are categorized as having more than one color.

As expected, when color is isolated as the only changing characteristic, it most significantly influences the visible region of the spectrum (400–700 nm). For instance, blue-colored objects exhibit a reflectance peak near 470 nm, while white-colored objects show consistently high reflectance across the visible range (**Figure 7b**). In the NIR–SWIR region, absorption features associated with polymer type remain unchanged regardless of color. When different

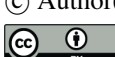

polymer types of the same color are compared, similar peaks exist in the visible region, as expected, with some minor
variations (**Figure 7c- d**).

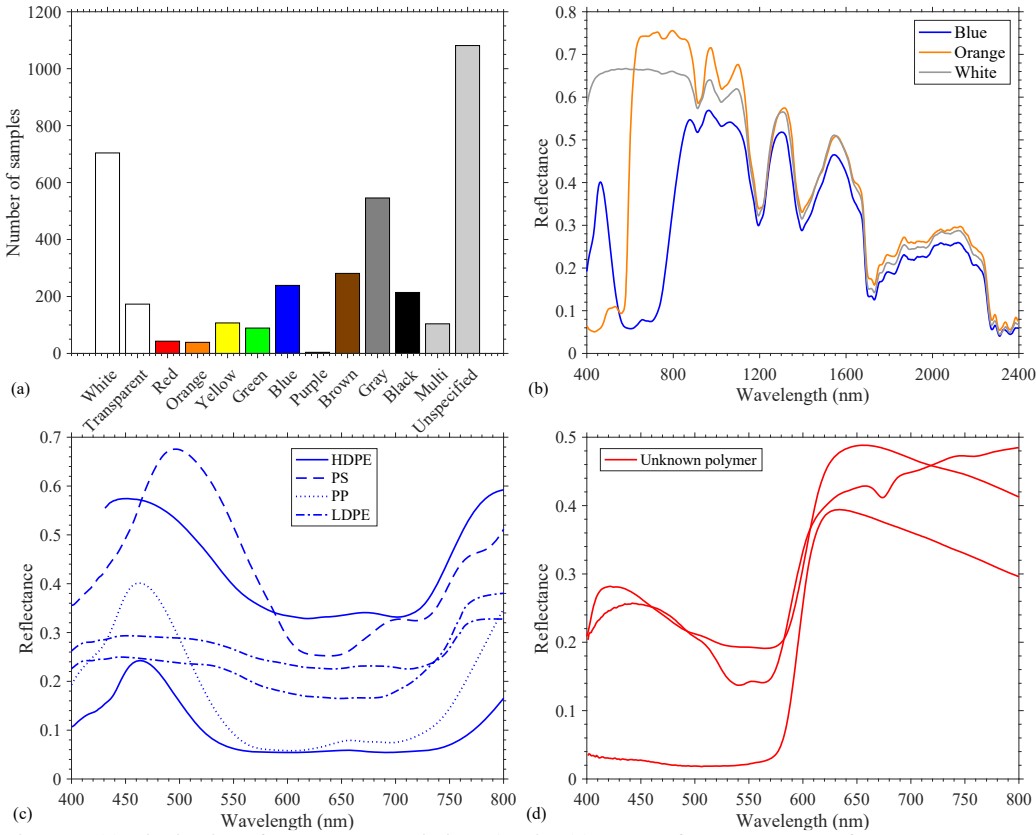

**Figure 7. (a) Distribution of sample colors within MADLib, (b) mean reflectance spectra of three polypropylene placemats in three different colors (c) six blue colored samples of different polymer types, and (d) three red samples of unknown polymer type.**


**3.4 Size**
Using the available quantitative and qualitative size data, all samples are categorized as macro, micro or uncategorized
due to a lack of available size data. Nearly half *(*41 %) of the samples within the dataset are unable to be categorized
(**Figure 8a**). Of those categorized, the majority (90 %) are considered macro-sized, and only 10 % are micro-sized.
Our results show that absorption features are consistent for micro- and macro-sized debris of the same polymer types
(**Figure 8b-d**). We note that there are differences in spectral features within the visible region, which are likely due
to color.

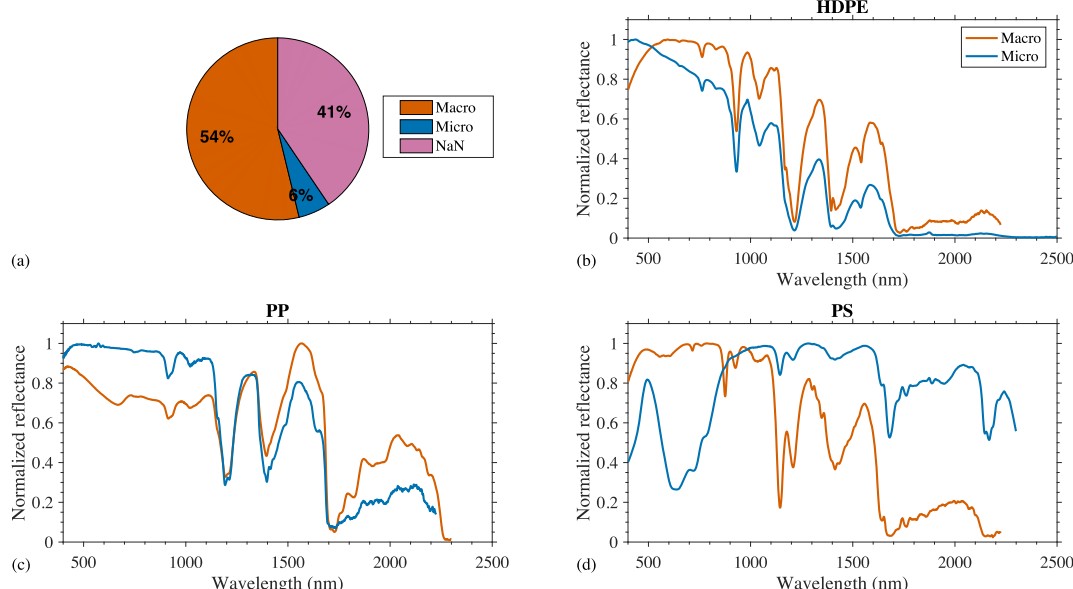

**Figure 8. (a) Categorical size distribution of samples within MADLib; and (b-d) representative mean reflectance spectra of micro- and macro-sized (b) HDPE (c) PP, and (d) PS. All reflectance spectra were normalized to their respective maximum values. Note: plotted micro- and macro-sized debris are from different datasets.**

## 3.5 Weathering state

The three categories of weathering - pristine, lab weathered and naturally weathered - have relatively equal contributions to the curated dataset, with less than one percent of samples left undefined (**Figure 9a**).

Case studies of three polymer types (PP, HDPE, and PA6) are presented before and after lab weathering (**Figure 9b-d**). Naturally weathered samples are not compared in the case study because no samples were measured before and after natural weathering for comparison. The two types of lab weathering, biofouling and UV exposure, produce different effects on reflectance within the visible spectrum. Samples exposed to UV radiation (dotted orange line) follow similar trends to their pristine counterparts with elevated reflectance values within the 1200-1600 nm range (**Figure 9b).** In comparison, all biofouled samples (solid orange lines) show reduced reflectance across the visible spectrum and exhibit a pronounced chlorophyll-a absorption feature at 670 nm (**Figure 9b–d**). No major differences are found in the NIR-SWIR region before and after biofouling; all the major spectral features for each polymer type remain the same.

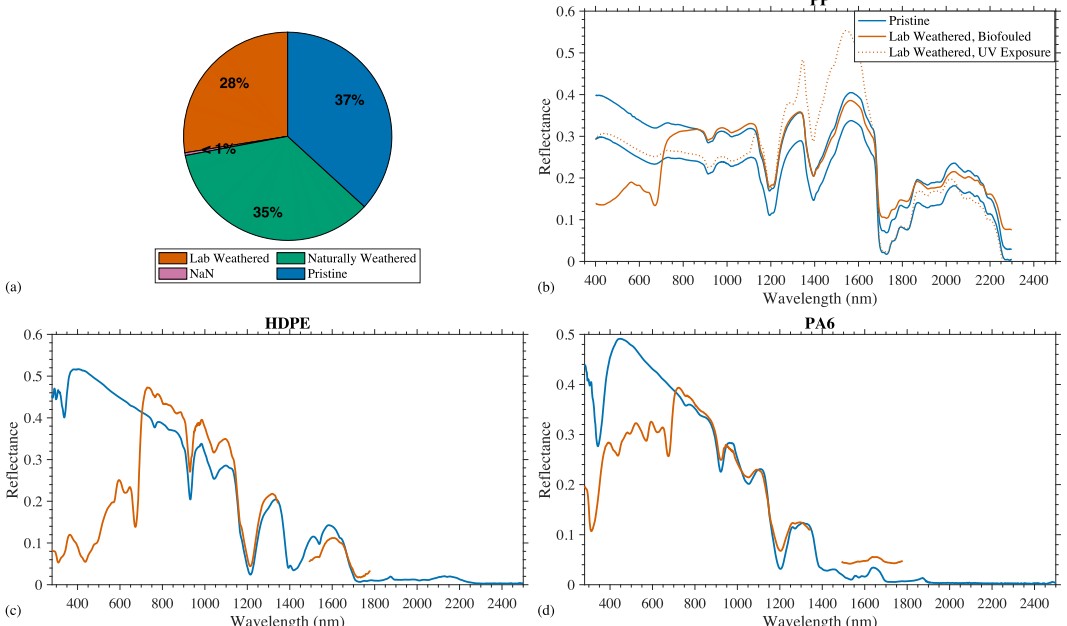

**Figure 9. (a) Distribution of weathering state categories within MADLib; and (b-d) representative mean reflectance spectra for pristine and lab-weathered (biofouled or UV-exposed) samples: (b) PP, (c) HDPE, and (d) PA6.**

### 3.6 Aqueous state

In MADLib, submerged samples constitute the largest aqueous state category (43 %), followed by dry samples (35 %), with fewer classified as floating (14 %) or wet (8 %) (**Figure 10a**). Each submerged object was measured at 3-20 separate depths to assess the effect of depth on spectral features, influencing the overall distribution. Less than one percent of samples is missing an aqueous state classification.

To examine the effect of the aqueous state on reflectance, a case study on the mean reflectance of pristine polypropylene samples from several datasets is presented. Reflectance magnitude decreases with increasing water interference in all cases, being highest for dry samples, followed by wet and floating/submerged samples. The case study reveals consistent spectral features in dry polypropylene across all four datasets (**Figure 10b-e**). The same spectral features are present for wet polypropylene samples as well (**Figure 10e-f**), but not present in submerged samples (**Figure 10c, e, f**). All submerged samples lose signal in the SWIR and their reflectance magnitudes decrease as depth increases (**Figure A1**), both of which are expected due to water's high absorption in the IR (Garaba and Dierssen, 2020). Submerged samples exhibit unique peaks at approximately 810 and 1070 nm, which are consistent across datasets (**Figure 10c, e, f**) and polymer types (**Figure A1**).



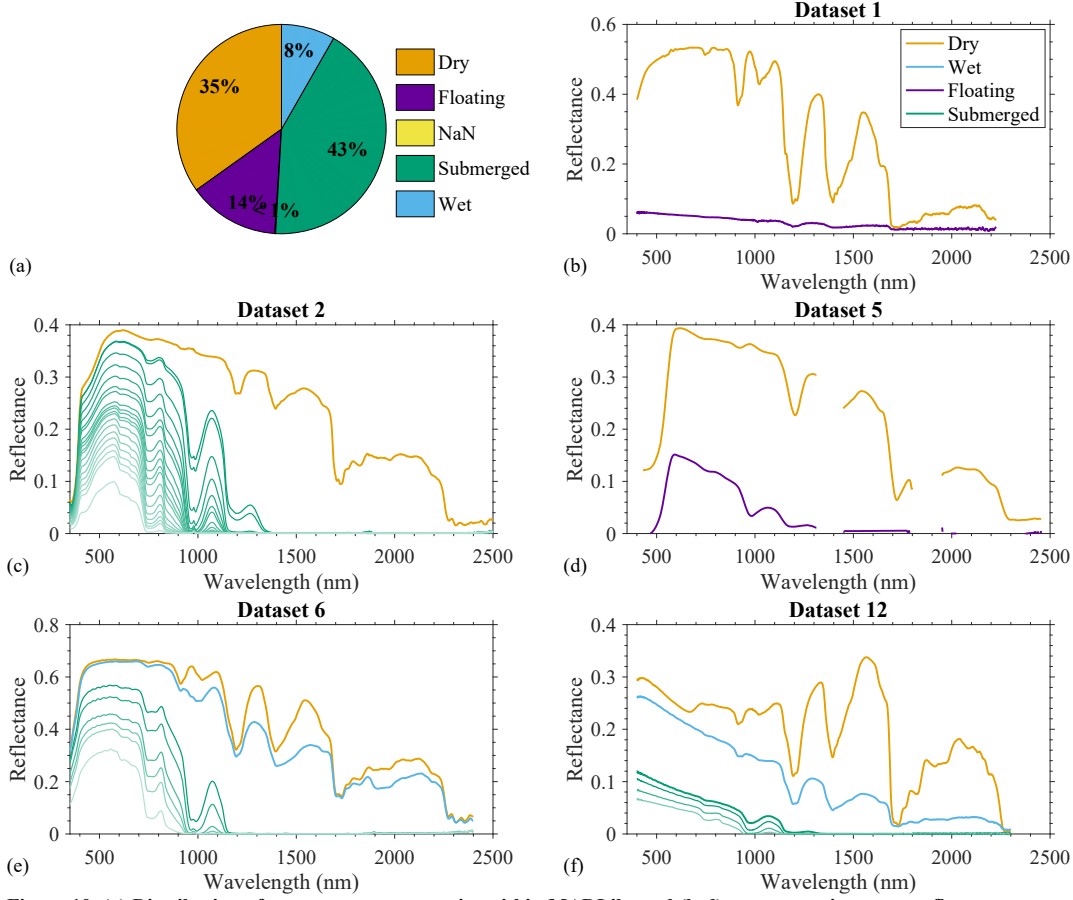

**Figure 10. (a) Distribution of aqueous state categories within MADLib; and (b-d) representative mean reflectance spectra**
**of dry polypropylene across four aqueous state categories, using datasets (b) 1, (c) 2, (d) 5, (e) 6, and (f) 12. Submerged**
**samples (c, e, f) are shown at depths ranging from 5 to 715 mm.**

## 4. Discussion

The analysis of MADLib revealed distinct spectral features linked to various characteristics of debris, as well as key
gaps in available data. Below, we interpret these findings, highlight MADLib's utility and limitations, and offer
recommendations for future data collection and research directions.

### 4.1 Preliminary lessons from MADLib

MADLib offers a valuable starting point for the development of algorithms aimed at detecting marine debris,
particularly plastics. Preliminary analyses highlight the role of color and biofilm presence on reflectance within the
visible spectrum (**Figure 7b, 9b**), whereas polymer type and aqueous state more strongly affect reflectance in the
SWIR region (**Figure 5b, 10b-f**). These findings support previous work (Knaeps et al., 2021; de Vries et al., 2023a).
Given the limited polymer-specific features within the visible range, we recommend focusing on SWIR wavelengths



for algorithm development. Notably, distinct spectral differences between dry and submerged plastics (**Figure 10c, e,**
**f**) suggest that separate detection algorithms may be required for optically bright terrestrial or dark.aquatic
environments.
Although MADLib includes many samples categorized under the same polymer type and aqueous state, no two
samples are identical. Every sample differs by at least one physical property or measurement condition, creating both
challenges and opportunities. On one hand, intra-category comparisons (e.g., among polypropylene samples) may be
confounded by variation within other sample characteristics or experimental design (**Figure 5c, d**). On the other hand,
this heterogeneity mirrors real-world conditions and provides a chance to identify robust spectral indicators that persist
across variability. In this way, MADLib functions as both a testing ground for existing algorithms (Asadzadeh and
Filho, 2017; Kühn et al., 2004; Zhang et al., 2022; Guo and Li, 2020; Garaba and Dierssen, 2018) and a platform for
developing new models that can handle spectral noise and natural diversity.
**4.2 Considerations for future work**
MADLib would benefit from more complete metadata and greater representation of common debris types to increase
its utility, as missing or inconsistent metadata currently hinders algorithm development. For example, the reflectance
spectrum of a plastic object (e.g., a dry cup) offers limited insight if essential metadata like polymer type, color, or
experimental design are missing. Many published datasets included in MADLib exhibited this issue. To address this,
we propose a comprehensive metadata structure (**Table B1**) for future marine debris reflectance studies. We sorted
potential metadata into "required", "best practice", and "as needed" categories, acknowledging that some metadata
maybe be difficult to obtain or unnecessary for future studies. **Table B2** summarizes the proposed changes to existing
parameters (color and size) and introduces a new parameter (object ID).
Future additions to MADLib should prioritize providing data on debris types that are currently underrepresented
within the collection. For example, polymers such as PS, PE, and PPA along with colors like yellow, green, brown,
and red have been recorded in marine debris surveys (Mutuku et al., 2024; Martí et al., 2020) but are poorly
documented in MADLib (**Figure 5a, 7a**). Furthermore, floating samples were rarely included in MADLib (**Figure**
**10a**) yet they are the most detectable type of marine debris via remote sensing, warranting their characterization in
future efforts. In addition, differences in the slopes of spectral features among samples of the same polymer type
highlight the need for further investigation into the causes of such variability (**Figure 5c-d**). Differences in
manufacturing processes, the presence of additives, and variations in experimental design are all potential factors that
could contribute to these discrepancies. Future work should examine the effects of additives and other manufacturing
treatments within the same polymer type, as these may influence optical properties and, by extension, detection and
classification accuracy.
While our initiative focused on reflectance, the most widely available optical parameter, future curation efforts that
incorporate additional optical properties will expand the applicability of MADLib across a broader range of sensors,



including active systems (Palombi et al., 2022; Goddijn-Murphy et al., 2024; Behrenfeld et al., 2023; de Fockert et
al., 2024).
MADLib also paves the way for collaborations among remote sensing scientists, modelers, and marine policy experts,
The current dataset may also support mapping debris movement and, when combined with physical transport models
e.g., (Maximenko and Hafner, 2024), can be used to forecast debris pathways to inform cleanup efforts and promote
polluter accountability. Lastly, we would like to emphasize the need for open-science and open-access approaches to
move this effort forward.

**Conclusions**

MADLib represents a foundational step toward harmonizing spectral reflectance measurements for marine debris and
is aligned with open-science policies. An important feature of the established MADLib collection is the traceable
curation that allows ingestion of data from any permanent repository, dataset or reference library (e.g., Ocean Scan,
PANGAEA, SeaBASS, SPECCHIO, USGS Spectral Library). We envision MADLib as a living resource where new
datasets can be added to maximize interoperability and findability of the collection. We believe that prioritizing the
measurements and metadata gaps discussed in future research will strengthen MADLib as a remote sensing community
resource. With its currently available data, and future iterations, MADLib will further support algorithm development
and help establish important specifications for debris detection to be implemented in future remote sensing
technologies.



**Appendix A**

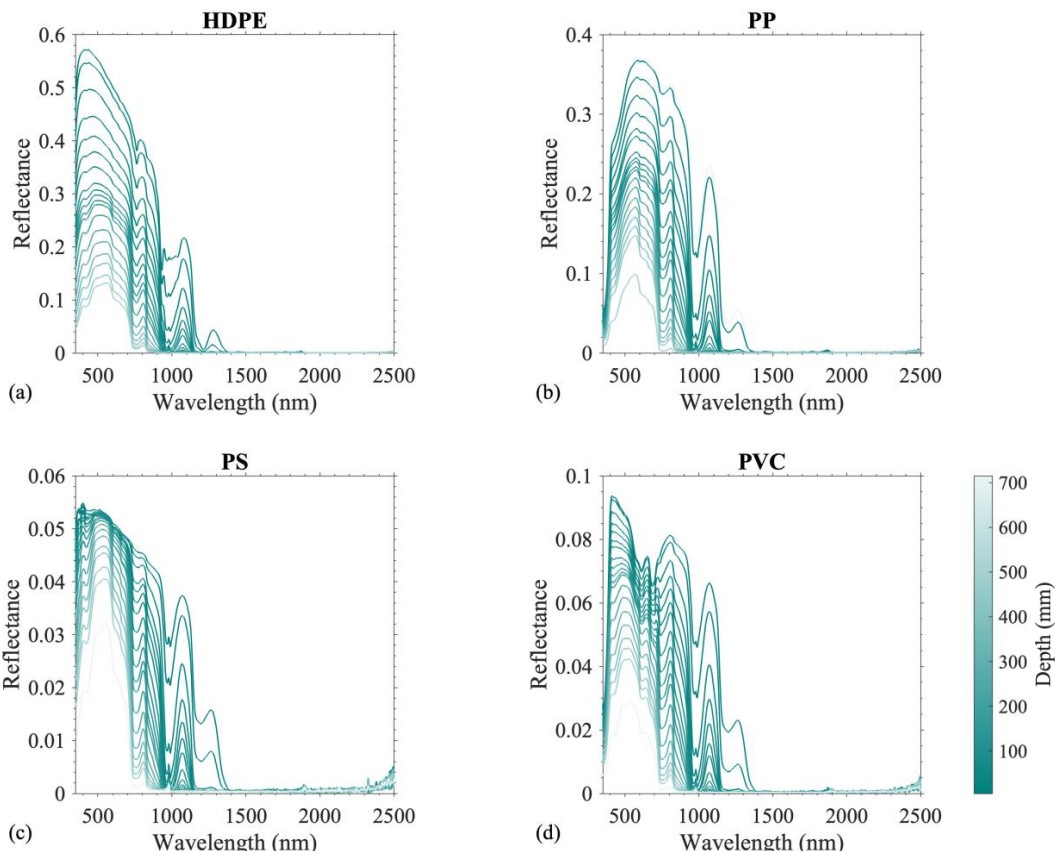

**Figure A1. Mean reflectance spectra of four polymer types – HDPE, PP, PS, and PVC - submerged to depths between 5-715 mm.**

**Appendix B**

**Table B1. Recommended metadata for future datasets, with required, best practice, and as needed metadata. See Table 2 for descriptions of metadata parameters.**

| Required | Best Practice | Optional |
|---|---|---|
| Object ID | Polymer Type | Lab Weathering Type |
| Object Type | Additives | Submergence Depth (mm) |
| Object State | | Water Type |



| | | |
|---|---|---|
| Origin | | TSM (mg/L) |
| Color | | Algal Cell Density (cells/mL) |
| Categorical Size | | Pixel Coverage (%) |
| Weathering State | | Other |
| Aqueous State | | |
| Instrument | | |
| Setting | | |
| Lighting | | |
| Background | | |
| Reference Standard | | |
| Fixed Height from Sample (m) | | |
| FoV (deg) | | |
| Viewing Geometry (deg) | | |

372

373  **Table B2. Recommended improvements for future metadata collection of specific descriptors.**

| Descriptor | Structure in MADLib | Proposed metadata information |
|---|---|---|
| **Color** | Uses "Multi" for objects with multiple colors as described by original authors | Instead of using "Multi," list each observed color to improve interpretation of reflectance in the visible spectrum |
| **Size** | Provides both categorical (e.g., micro, macro) and dimensional (length, width, height) information | Categorical labels are sufficient if consistent cutoffs are applied (e.g., micro < 5 mm; macro ≥ 5 mm), as size had minimal impact on spectral feature locations (**Figure 5b**) |
| **Object ID** | Not currently included | Add to identify the same object measured under different conditions (e.g., dry vs. submerged) |

374



**Author contribution**

Conceptualization: AO, KB, SPG; Data curation: AO; Formal Analysis: AO; Supervision: KB, SPG, SRC; Funding Acquisition: KB, SRC; Writing – original draft preparation: AO; Writing- Review and Editing: SG, KB, SRC. All the authors reviewed and approved the manuscript text.

**Competing Interests**

The authors declare that they have no conflict of interest.

**Acknowledgements**

The study was supported as part of a summer internship by AO at Science Mission Directorate at the National Aeronautics and Space Administration headquarters. We appreciate the feedback on the study from members of Rivero-Calle laboratory, Jay Brandes and Daniel Koestner. We thank Lee Ann Deleo for her help with Figure 1. Finally, we thank the authors who contributed to the original datasets (Corbari, de Vries, English, Garaba, Leone, and co-authors) without which this work would not be possible.

**Financial Support**

AO was supported by the University of Georgia Graduate Tuition Return Incentive Program (GTRIP, 2023-2025) and 2024 National Aeronautics and Space Administration Headquarters summer internship reference code – 016758. SPG was supported through Deutsche Forschungsgemeinschaft grant no. 417276871 and Basic Activities contract no. 4000132037/20/NL/GLC within the Discovery Element of the European Space Agency.

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
