# Peer review of "The MArine Debris hyperspectral reference Library collection (MADLib)"

_Earth System Science Data, 2025_

## Author Comment (AC1)

**Reviewer 1 comments**

The manuscript addresses an important topic, compiling a comprehensive dataset of mostly plastic litter reflectance spectra into a spectral library for use in plastic litter detection algorithm development and assessment. The structure is generally good, the writing is clear, and the figures are helpful in presenting results. However, some aspects require clarification and minor additions to improve transparency and interpretation of results, specifically with regard to spectral analysis and interpretation.

We thank the reviewer for taking the time to provide constructive feedback and suggestions to improve the manuscript. Below, each comment has been carefully addressed with specific details on how and where revisions were added, with additional text shown in *italics*.

C1: A more thorough comparison between datasets for the same samples would be useful. When significant variation is present, this should be examined and reported. See comment on figure 5 below.

R1: Thank you for this comment, we agree that addressing variability within and among datasets is important. We added details comparing between datasets where relevant, both in section 3.1 and 3.4. In section 3.1 (line 244), the following text was added: "The spectral shape in the visible spectrum is consistent with the apparent color of each object; this is further explored in Section 3.3. In the case of the PP samples, the objects in dataset 6 were observed to be transparent, orange, and blue in color, while those in datasets 1 and 7 were white/transparent, those in dataset 2 were grey, and those in dataset 5 were yellow and brown (Figure 5d). Dataset 12 did not provide information on apparent color. The absorption features within the SWIR are consistent across samples of the same polymer type and align closely with reported literature…"

We also added new text to section 3.4 (line 278): "Our results show that the locations of absorption features within the SWIR are consistent for micro- and macro-sized debris of the same polymer types (Figure 8b-d). In each example, the microplastic and macro-plastic measurements are from different datasets (Figure 8b-d). Therefore, there are differences in sample color as well as experimental setup (such as field of view or height of spectroradiometer from sample) between the two samples. Differences in spectral features within the visible region are likely due to color inferred from the corresponding peaks and shapes. Other slight differences in feature location may be attributed to other metadata (e.g. additives, state of weathering, etc.)."

C2: Comparison of findings from this study to other studies assessing the same effects of weathering and submersion on reflectance can be improved.

R2: We revised our discussion section 4.1 to provide a clearer comparison of our findings with respect to other studies, with the following new text (line 337): "Previous studies have also found apparent color to influence spectral shape and magnitude in the visible spectrum, while the SWIR region remains largely unaffected regardless of the intrinsic color of the object (Knaeps et al., 2021; Garaba et al., 2021a). Biofouling of plastics has similar results, with the

SWIR properties remaining largely unchanged, while spectral shape and magnitude are altered in the visible spectrum exhibiting a pronounced red edge feature (de Vries et al., 2023b). Measurements of wet, floating, and submerged debris show reduction in magnitude across the spectrum, especially in the SWIR, compared to dry debris (Knaeps et al., 2021; Garaba et al., 2021a; de Vries et al., 2023b). Submerged objects have minimal signal in the NIR-SWIR compared other aqueous states and unique spectral features in the NIR near 810 and 1070 nm (Knaeps et al., 2021; de Vries et al., 2023b). Distinct spectral differences between submerged plastics and other aqueous states (Figure 10c, e, f) suggest that separate detection algorithms may be required for submerged debris using the visible or NIR spectral regions. Given the limited polymer-specific features within the visible range, we further recommend focusing on SWIR wavelengths for algorithm development of non-submerged debris as prior studies have suggested (Garaba and Harmel, 2022)."

C3: Most current sensors lack the capacity of distinguishing narrow SWIR features. The authors state that SWIR should be used for algorithm development. How would that be put in practice? Are the authors referring specifically to VHR hyperspectral platforms?

R3: We appreciate this feedback and great point about distinguishing SWIR features. The discussion section 4.1 was updated with examples of studies that have accomplished debris detection with SWIR features and the tools needed to do so (line 349): "Several studies have demonstrated prospective plastic detection using high-resolution sensors with appropriate SWIR spectral bands (Guo and Li, 2020; Asadzadeh and Filho, 2016; Park et al., 2021; Kremezi et al., 2021), suggesting that prioritizing future sensors with appropriate spatial resolution (e.g. 30 m or less) and band placement can make SWIR-based approaches viable."

C4: Additionally, most polymers exhibit similar absorption features, which are characteristic of plastics in general, which we can also see from figure 5 in this study. How useful and/or practical is it to be looking at/for specific polymer types? Would a mean spectrum for dry, wet, submerged, biofouled, weathered etc. spectrum for plastics in general be useful for algorithm development?

R4: While absorption features seem very similar across all polymers, there have been several publications that have started to look into polymer separation (Castagna et al., 2023; Masoumi et al., 2012; Moroni et al., 2015) based on small differences in these features. MADLib can be used to test existing algorithms and continue to develop new ones to classify polymers based on spectral reflectances. We added the following text to section 4.1 of the manuscript to address this comment (line 358): "In this way, the breadth of measurement conditions available in MADLib can serve as a testing ground for existing plastic detection algorithms (Asadzadeh and Filho, 2017; Kühn et al., 2004; Zhang et al., 2022; Guo and Li, 2020; Garaba and Dierssen, 2018) and to derive general plastic endmember spectra (Hu, 2025). Beyond general plastic detection, it also provides opportunities for developing new approaches, including testing the feasibility of polymer-specific identification algorithms (Castagna et al., 2023; Moroni et al., 2015; Masoumi

et al., 2012; Huth-Fehre et al., 1995). Detection of specific polymer types would aid in cleanup initiatives, polluter accountability, and policy development (NASEM, 2021)."

**Specific comments:**

C5: Line 142: Shouldn't atmospheric absorption only matter when dealing with dense layers of the atmosphere? Since measurements are taken with instrument at less than a meter from sample, instrument noise can be present but atmospheric effects should be negligible. As a matter of fact, this is the reason we usually don't perform any atmospheric correction when dealing with UAV data. And this is where these spectra areas would come in handy.

R5: We thank the reviewer for mentioning this point of confusion. Atmospheric absorption was only present in outdoor measurements; an updated explanation of this has been added to section 2.3.4 (line 144): "Strong atmospheric absorption occurs *in the SWIR*, specifically around 1350–1450 nm, 1800–1950 nm, and above 2400 nm (Garaba and Dierssen, 2020; Clark et al., 2003). This effect is observed *in outdoor measurements*, where sunlight interacts with larger portions of the atmosphere before reaching the sample. Spectra often exhibit abrupt, isolated peaks due to these atmospheric absorption bands."

C6: 2.4.4 weathering: Besides the type of weathering, the degree of weathering can also affect reflectance properties significantly. Perhaps authors should consider adding a degree of weathering metadata field, or state that all samples were similarly weathered if that is the case.

R6: We agree that the degree of weathering is a great point to add to the metadata. This recommendation has been added to the 'optional' column in appendix table B1 for future dataset use, as well as to table B2.

C7: 2.4.5 aqueous state: I can see from the metadata file that the depth of submersion is reported. This should be reflected in the text.

R7: Agreed. The following was added to the end of section 2.4.5 (line 212): "Additionally, the submergence depth of samples in millimeters under the water surface was provided in the submergence depth (mm) column, ranging from 5-715 mm."

C8: Lines 216-218: Does pixel coverage affect reflectance spectra magnitude? Have you taken this into account when presenting spectra?

R8: Yes, pixel coverage does impact reflectance magnitude. This is accounted for when samples were selected for comparison in figures. Below are additions to the text to make sure this is clear:

- Response 9 below addresses pixel coverage for figure 5/section 3.1.
- Added to section 3.4 (line 284): "Additionally, we note that the HDPE microplastic spectrum has a pixel coverage of 90%, compared to 100% for all other spectra."
- Response 12 below addresses pixel coverage for figure 10/section 3.6.

- C9: Lines 218-219: Same with the glass, have you examined how this could affect measurements and when presenting spectra?
- R9: To address the impact that glass presence may have on magnitude, the following was updated in section 3.1 (line 250): "The observed differences in magnitude between spectra from a single dataset are associated with changes in sample thickness and size (datasets 1,2, and 12), glass presence (dataset 2), or pixel coverage (dataset 5)."
- C10: Figure 5: mean reflectance spectra of HDPE (5c) and PP (5d) samples present some pronounced variation between datasets, specifically in the visible range. Is this due to different colours? If so this should be reflected in the text. I can see that the authors touch on this in the discussion section. However as the differences are quite pronounced, I believe this warrants further examination or explanation and should be given more weight in the results section as well. If this is not due to colour, can authors hypothesize as to what it might be? Could it mean there are measurement validity issues?
- R10: Thank you for pointing this out and it is indeed an effect of color on the visible spectrum. In section 3.1 (line 244), the following text was added: "The spectral shape in the visible spectrum is consistent with the apparent color of each object; this is further explored in Section 3.3. In the case of the PP samples, the objects in dataset 6 were observed to be transparent, orange, and blue in color, while those in datasets 1 and 7 were white/transparent, those in dataset 2 were grey, and those in dataset 5 were yellow and brown (Figure 5d). Dataset 12 did not provide information on apparent color. The absorption features within the SWIR are consistent across samples of the same polymer type and align closely with reported literature...".
- C11: Figure 5: Normalization methodology could be better defined.
- R11: We have revised the caption to better explain the normalization method, captions of all normalized figures (Figures 5 and 8) were changed to: "Each reflectance spectrum was normalized to its own maximum reflectance value."
- C12: Figure 10: There is a significant variation in reflectance magnitude of floating PP samples between dataset 1 (figure 10b) and dataset 5 (figure 10d) in the visible part of the spectrum. Have authors examined why this is the case?
- R12: For more thorough examination of the floating samples, a figure of just the floating samples has been added to the Supplement (Figure A2). The differences of the visible spectrum are due to object color. Additional explanation has been added to section 3.6 (line 317): "The floating objects represented from datasets 1 and 5 have pixel coverages of 66% and 60%, respectively, representing the highest available pixel coverage for dataset 1 and the closest match for dataset 5 (Figure 10b, d). The dataset 1 sample exhibits the same prominent SWIR absorption features when dry and wet, while the reflectance shape of the dataset 5 floating sample more closely resembles a submerged debris item (Figure A2). The reflectance shape in the visible region is

related to the color of the object in both cases. Both floating measurements have more noise and missing values in the infrared than the dry measurements."

C13: Lines 312-313: Aqueous state affects reflectance magnitude across the full range of spectrum, both in the NIR/SWIR and visible (although less pronounced). If authors here mean the shape of the spectrum this should be clearly stated.

R13: The specification of reflectance shape has been added (line 335): "Preliminary analyses highlight the role of *apparent* color and biofilm presence on *spectral shape in* the visible spectrum (Figure 7b, 9b), whereas polymer type and aqueous state more strongly affect *spectral shape* in the SWIR region (Figure 5b, 10b-f)."

**Technical corrections:**

C14: Data availability is missing section numbering

R14: The section number has been added, thank you (section 5).

C15: Figure 1 resolution should be improved

R15: We re-uploaded this figure with higher resolution.

C16: Table 2 formatting for line stroke

R16: Thank you, we have fixed the table.

C17: Conclusions is missing section numbering

R17: Thank you, we numbered it (section 6).

---

## Author Comment (AC2)

**Reviewer 2 Comments**

The MS is well organized, focusing on a relevant issue concerning marine litter detection. The idea to create a spectral library, facing the main issue connected with laboratory experiments, is essential. The MS is well structured and the writing is clear.

We thank the reviewer for the positive feedback on our manuscript. Below, each comment has been carefully addressed with specific details on how and where revisions were added, with additional text shown in *italics*.

Following are reported some suggestions:

C1: Figure 5b is not clear; is it regards the meac od the spectra among all datasets?

R1: Data within MADLib are available as means, medians, and standard deviations of the replicate measurements for each sample. The following was added to the beginning of the results for clarity (line 231): "To ensure consistency, all spectra plotted in this study correspond to the mean of replicate measurements for each sample."

The caption for Figure 5b was updated to: "(b) example reflectance spectra for each polymer type in MADLib".

C2: Figure 5d: why the authors refer to PP samples? does it have a different signature than the others?

R2: Figures 5c and 5d are example plots of two polymer types in MADLib to demonstrate intravariability for polymers. An explanation for polymer selection was added to section 3.1 (line 242): "The reflectance spectra of all pristine, dry PP and HDPE samples are isolated and presented separately to inspect intra-variability in spectral features of measured polymers (Figure 5c, d). These polymers were selected because they are well represented in MADLib, with numerous samples originating from several independent datasets."

C3: Figure 5: the spectra compared are acquired in the same way?

R3: : Good point, all of the compared spectra come from the different datasets included in MADLib. We added the following phrase in section 3.1 to clarify this (line 252): "Measurements between datasets also differed in acquisition, such as fiber optic field of view and height of instrument from sample at time of measurement, which may affect pixel coverage."

C4: Figure 7: the spectra are referring to the same polymer having different color?

R4: Yes, Figure 7b refers to the same polymer (polypropylene) in three different colors. Figure 7b caption was updated to "reflectance spectra of the same polymer and object (polypropylene placemat) in three different colors".

C5: For the abstract and introduction, I was expecting to read more information about the protocol to be used to acquire the spectra via a laboratory experiment. However, no information are reported regarding the different techniques and the differences in spectral data.

R5: We agree that the protocols used to gather the data are important. We believe this addition fits best in methods section 2.4.6 (line 225) rather than the introduction: "Further information on protocols specific to each dataset may be found in the original publications (Corbari et al., 2024; Corbari et al., 2020; de Vries et al., 2023a; English and Hu, 2020; Garaba and Dierssen, 2020; Knaeps et al., 2021; Garaba et al., 2021a; Garaba et al., 2021b; Leone et al., 2023)."

Additionally in the introduction, line 62 was updated: "However, current marine debris reflectance datasets have been gathered using variable methodologies, data processing, and metadata curation, making it challenging to combine them for algorithm development and to identify gaps in the field."